# Translocation of interleukin-1β into a vesicle intermediate in autophagy-mediated secretion

**Min Zhang[1], Samuel J Kenny[2], Liang Ge[1], Ke Xu[2], Randy Schekman[1]***

[1]Department of Molecular and Cell Biology, Howard Hughes Medical Institute, University of California, Berkeley, Berkeley, United States; [2]Department of Chemistry, University of California, Berkeley, Berkeley, United States

**Abstract** Recent evidence suggests that autophagy facilitates the unconventional secretion of the pro-inflammatory cytokine interleukin 1β (IL-1β). Here, we reconstituted an autophagy-regulated secretion of mature IL-1β (m-IL-1β) in non-macrophage cells. We found that cytoplasmic IL-1β associates with the autophagosome and m-IL-1β enters into the lumen of a vesicle intermediate but not into the cytoplasmic interior formed by engulfment of the autophagic membrane. In advance of secretion, m-IL-1β appears to be translocated across a membrane in an event that may require m-IL-1β to be unfolded or remain conformationally flexible and is dependent on two KFERQ-like motifs essential for the association of IL-1β with HSP90. A vesicle, possibly a precursor of the phagophore, contains translocated m-IL-1β and later turns into an autophagosome in which m-IL-1β resides within the intermembrane space of the double-membrane structure. Completion of IL-1β secretion requires Golgi reassembly and stacking proteins (GRASPs) and multi-vesicular body (MVB) formation.

***For correspondence:**
schekman@berkeley.edu

## Introduction

Most eukaryotic secretory proteins with an N-terminal signal peptide are delivered through the classical secretion pathway involving an endoplasmic reticulum (ER)-to-Golgi apparatus itinerary (*Lee et al., 2004*; *Schatz and Dobberstein, 1996*). However, a substantial number of secretory proteins lack a classical signal peptide, called leaderless cargoes, and are released by unconventional means of secretion (*Nickel and Rabouille, 2009*; *Nickel and Seedorf, 2008*). The range of unconventional secretory cargoes encompasses angiogenic growth factors, inflammatory cytokines and extracellular matrix components etc. most of which play essential roles for development, immune surveillance and tissue organization (*Nickel, 2003*; *Rabouille et al., 2012*). Unlike a unified route for classical protein secretion, leaderless cargoes undergoing unconventional secretion employ multiple means of protein delivery, the details of which are largely unknown (*Ding et al., 2012*; *Nickel, 2010*; *Rabouille et al., 2012*; *Zhang and Schekman, 2013*).

IL-1β is one of the most intensely investigated cargoes of unconventional secretion. A biologically inactive 31 kDa precursor, pro-IL-1β, is made following initiation of the NF-κB signaling cascade. Pro-IL-1β is subsequently converted into the active form, the 17 kDa mature IL-1β, by the pro-inflammatory protease caspase-1 which is activated, in response to extracellular stimuli, after its recruitment to a multi-protein complex called the inflammasome (*Burns et al., 2003*; *Cerretti et al., 1992*; *Rathinam et al., 2012*; *Thornberry et al., 1992*). Interpretation of the mechanism of unconventional secretion of IL-1β is complicated by the fact that one of the physiologic reservoirs of this cytokine, macrophages, undergoes pyroptotic death and cell lysis under conditions of inflammasome activation of caspase-1. Indeed, many reports including two recent publications make the case for cell lysis

**eLife digest** Cells release a large number of proteins to the extracellular space. The majority of these 'secreted' proteins first pass through two structures inside cells called the endoplasmic reticulum and Golgi. However, a growing number of proteins have been identified that are released by an unconventional mechanism that bypasses the endoplasmic reticulum and Golgi.

Autophagy is a process that destroys damaged proteins and other unwanted material in cells. It gets triggered when cells are starved of nutrients, leading them to digest their own materials and recycle the resources into new molecules. During autophagy, a cup-like structure with a double layer of membrane forms around the material that is to be digested. This structure then elongates and eventually engulfs the material to form a bubble-like compartment called the autophagosome. Recent evidence has suggested that autophagosomes are involved in the unconventional secretion of a protein called interleukin-1β; this protein is crucial for the body's immune response against infection. However, it was not clear how these proteins entered the autophagosomes.

Zhang et al. have now explored the link between interleukin-1β and autophagy in more detail. The experiments showed that when autophagy was triggered by starvation, the secretion of interleukin-1β was enhanced. Conversely, when autophagy was inhibited, interleukin-1β accumulated inside the cells and could not be secreted.

Further experiments then revealed unexpectedly that interleukin-1β was not engulfed by the cup-like structure (as is the case for material that is destined to be removed). Instead, interleukin-1β was found to enter into smaller bubble-like packages (called vesicles) that turn into the autophagosome.

Zhang et al. also found that a protein called HSP90 binds to interleukin-1β and enables it to cross the membrane (or translocate) into the vesicles, and that this means that interleukin-1β actually resides in the space between the outer and inner membranes of the autophagosome. How many other proteins share this unusual route out of the cell and what membrane channel is used for this translocation event remain open questions for the future.

as a means of release of mature IL-1β (*Liu et al., 2014*; *Shirasaki et al., 2014*). In contrast, other reports demonstrate proper secretion of mature IL-1β without cell lysis in, for example, neutrophils, which are nonetheless dependent on the inflammasome response to activate caspase-1 and secrete mature IL-1β (*Chen et al., 2014*).

Quite aside from the possible complication of cell lysis, another body of work has suggested an unconventional pathway for the proper secretion of IL-1β. Pro-IL-1β lacks a typical signal peptide and the propeptide is processed in the cytosol rather than the ER (*Rubartelli et al., 1990*; *Singer et al., 1988*). Although mature IL-1β appears to be incorporated into a vesicular transport system, secretion is not blocked by Brefeldin A, a drug that blocks the traffic of standard secretory proteins form the Golgi apparatus (*Rubartelli et al., 1990*). Multiple mechanisms have been implicated in the unconventional secretion of IL-1β, including autophagy, secretory lysosomes, multi-vesicular body (MVB) formation and micro-vesicle shedding (*Andrei et al., 1999*; *Andrei et al., 2004*; *Brough et al., 2003*; *Lopez-Castejon and Brough, 2011*; *MacKenzie et al., 2001*; *Qu et al., 2007*; *Verhoef et al., 2003*). However, a clear demonstration of the mechanism for the entry of IL-1β into a vesicular carrier, e.g. the autophagosome, is lacking.

Macroautophagy (hereafter autophagy) is a fundamental mechanism for bulk turnover of intracellular components in response to stresses such as starvation, oxidative stress and pathogen invasion (*Mizushima and Levine, 2010*; *Yang and Klionsky, 2010*). The process is characterized by the formation of a double-membrane vesicle, called the autophagosome, through the elongation and closure of a cup-shaped membrane precursor, termed the phagophore, to engulf cytoplasmic cargoes (*Hamasaki et al., 2013*; *Lamb et al., 2013*). Completion of autophagosome formation requires a sophisticated protein-vesicle network organized by autophagic factors, such as autophagy-related (ATG) proteins, and target membranes (*Feng et al., 2014*; *Mizushima et al., 2011*). Besides the degradative function, autophagy or ATG proteins have recently been implicated in multiple secretory pathways including the delivery of leaderless cargoes undergoing unconventional secretion, such as the mammalian pro-inflammatory cytokines IL-1β and IL-18, the nuclear factor HMGB1, and the yeast acyl coenzyme A-binding protein Acb1, to the extracellular space (*Bruns et al., 2011*;

*Dupont et al., 2011*; *Duran et al., 2010*; *Manjithaya and Subramani, 2011*; *Pfeffer, 2010*; *Subramani and Malhotra, 2013*). The Golgi reassembly and stacking protein(s) GRASP(s) (GRASP55 and GRASP65 in mammals, dGRASP in *Drosophila*, GrpA in Dictyostelium and Grh1 in yeast) are required for autophagy-regulated unconventional secretion (*Giuliani et al., 2011*; *Kinseth et al., 2007*; *Levi and Glick, 2007*; *Manjithaya et al., 2010*).

*Dupont et al., 2011* documented a role for autophagy in the secretion of mature IL-1β (*Dupont et al., 2011*), but how a protein sequestered within an autophagosome could be exported as a soluble protein was unexplained. Here, we sought to understand how conditions of starvation-induced autophagy could localize IL-1β into an autophagosomal membrane. We reconstituted the autophagy-regulated secretion of IL-1β in cultured cell lines and detected a vesicle intermediate, possibly an autophagosome precursor, containing mature IL-1β. Three-dimensional (3D) Stochastic Optical Reconstruction Microscopy (STORM) demonstrated that, after entering into the autophagosome, IL-1β colocalizes with LC3 on the autophagosomal membrane, which, together with an antibody accessibility assay and observations from biochemical assays, implies a topological distribution in the intermembrane space of the autophagosome. This distribution of IL-1β explains the mechanism accounting for its secretion as a soluble protein through either a direct fusion of autophagosome with the plasma membrane or via the MVB pathway.

## Results

### Reconstitution of autophagy-regulated IL-1β secretion

A dual effect of autophagy has been proposed on the secretion of IL-1β in macrophages (*Deretic et al., 2012*; *Jiang et al., 2013*). On one hand, induction of autophagy directly promotes IL-1β secretion after inflammasome activation by incorporating it into the autophagosomal carrier (*Dupont et al., 2011*). On the other hand, autophagy indirectly dampens IL-1β secretion by degrading components of the inflammasome as well as reducing endogenous triggers for inflammasome assembly, including reactive oxygen species (ROS) and damaged components, which are required for the activation of caspase-1 and the production of active IL-1β (*Harris et al., 2011*; *Nakahira et al., 2011*; *Shi et al., 2012*; *Zhou et al., 2011*).

To focus our study specifically on the role of autophagy in IL-1β secretion, we reconstituted a stage of IL-1β secretion downstream of inflammasome activation by co-expressing pro-IL-1β (p-IL-1β) and pro-caspase-1 (p-caspase-1) in non-macrophage cells. As shown in *Figure 1A*, the generation and secretion (~5%) of mature IL-1β (m-IL-1β) was achieved by co-expression of p-IL-1β and p-caspase-1 in HEK293T cells. Mature IL-1β was not produced or secreted without p-caspase-1, whereas a low level of secreted p-IL-1β (~0.2%) was detected with or without the expression of p-caspase-1. Furthermore, little cell lysis occurred during the treatment we used to induce IL-1β secretion: Much less precursor than mature IL-1β and little cytoplasmic tubulin was detected released into the cell supernatant during the 2 hr incubation in starvation medium (*Figure 1A*). Starvation, a condition that stimulates autophagy, enhanced IL-1β secretion (~3 fold) and reduced the level of IL-1β in the cell lysates (*Figure 1A,B*). Inhibition of autophagy by the phosphatidylinositol 3-kinase (PI3K) inhibitors 3-methyladenine (3-MA) or wortmannin (Wtm) blocked IL-1β secretion activated by starvation and caused the accumulation of mature IL-1β in the cell (*Figure 1B*). Likewise, in an autophagy-deficient cell line, Atg5 knockout (KO) mouse embryo fibroblasts (MEFs) (*Mizushima et al., 2001*), IL-1β secretion was reduced and failed to respond to starvation (*Figure 1C*). Moreover, IL-1β secretion was also inhibited in a dose-dependent manner in the presence of an ATG4B mutant (C74A) (*Fujita et al., 2008*), or after the depletion of ATG2A and B (*Velikkakath et al., 2012*), or FIP200 (*Hara et al., 2008*), which block autophagosome biogenesis at different stages (*Figure 1D–F*). Therefore, the reconstituted system recapitulates the autophagy-regulated secretion of IL-1β.

In macrophages, MVB formation and GRASP proteins are required for IL-1β secretion (*Dupont et al., 2011*; *Qu et al., 2007*). Inhibiting MVB formation by depletion of the ESCRT components, hepatocyte growth factor receptor substrate (Hrs) or TSG101, compromised secretion of IL-1β and CD63, an exosome marker (*Figure 1—figure supplement 1A*). Knockdown of the GRASP55 or GRASP65 also led to the reduction of IL-1β secretion (*Figure 1—figure supplement 1B*). Therefore, in addition to functions required for autophagy, the secretion of IL-1β in HEK293T cells depends on

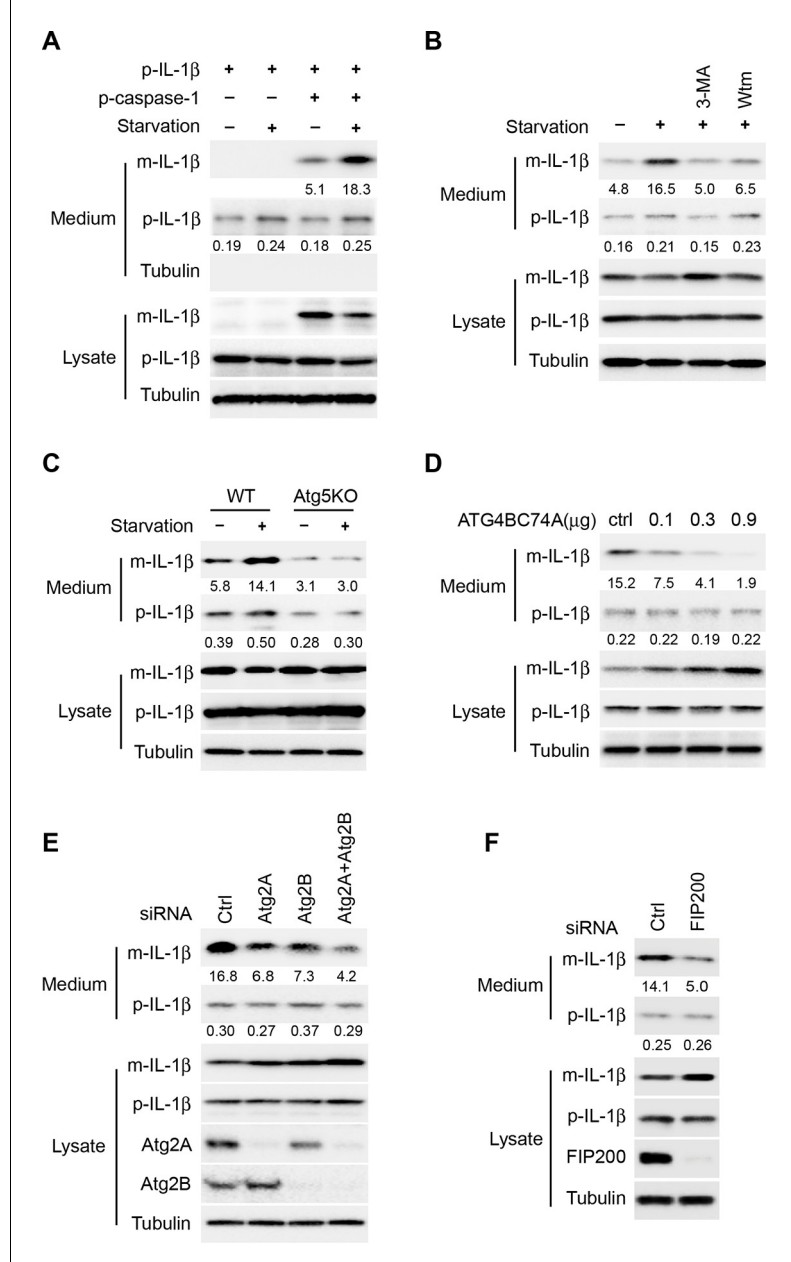

**Figure 1.** Reconstitution of autophagy-regulated IL-1β secretion in cultured cells. (**A**) Reconstitution of starvation-induced IL-1β secretion in HEK293T cells. HEK293T cells were transfected with a single plasmid encoding p-IL-1β or together with the p-caspase-1 plasmid. After transfection (24 h), the cells were either treated in regular (DMEM) or starvation (EBSS) medium for 2 hr. The medium and cells were collected separately and immunoblot was performed to determine the level of indicated proteins. (**B**) PI3K inhibitors 3-methyladenine (3-MA) or wortmannin (Wtm) inhibit IL-1β secretion. HEK293T cells transfected with p-IL-1β and p-caspase-1 plasmids were cultured in DMEM, EBSS, or EBSS containing 10 mM 3-MA or 20 nM wortmannin for 2 hr. The medium and cells were collected separately and immunoblot was performed as shown in (**A**). (**C**) IL-1β secretion is blocked in Atg5 KO MEFs. Control WT or Atg5 KO MEFs were transfected with p-IL-1β and p-caspase-1 plasmids. After transfection (24 hr), the cells were either cultured in DMEM or EBSS for 2 hr followed by immunoblot as shown in (**A**). (**D**) IL-1β secretion is inhibited by the ATG4B mutant (C74A). HEK293T cells were transfected with plasmids encoding p-IL-1β, p-caspase-1 and different amounts of ATG4B (C74A) plasmid DNA as indicated. After transfection (24 hr), cells were starved in EBSS for 2 hr followed by immunoblot as shown in (**A**). (**E**) Knockdown of Atg2 reduces IL-1β secretion. HEK293T cells were transfected with control siRNA or siRNAs against Atg2A, Atg2B alone or both. After transfection (48 hr), the cells were transfected with p-IL-1β and p-caspase-1 plasmids. After another 24 hr, the cells

*Figure 1. continued on next page*

*Figure 1. Continued*

were starved in EBSS for 2 h followed by immunoblot as shown in (**A**). (**F**) Knockdown of FIP200 reduces IL-1β secretion. HEK293T cells were transfected with control siRNA or FIP200 siRNA. IL-1β secretion under starvation conditions was determined as shown in (**E**). Quantification of IL-1β secretion was calculated as the ratio between the amount of IL-1β in the medium and the total amount (the sum of IL-1β in both medium and lysate).

The following figure supplement is available for figure 1:

**Figure supplement 1.** Depletion of ESCRT or GRASPs affects IL-1*β* secretion.

GRASP proteins and at least two proteins implicated in MVB formation, as reported previously (*Dupont et al., 2011*; *Qu et al., 2007*).

## IL-1β transits through an autophagosomal carrier during secretion

To study if autophagy directly regulates IL-1β secretion, we employed a three-step membrane fractionation procedure as described previously (*Figure 2A*) (*Ge et al., 2013*). We first performed a differential centrifugation to obtain 3k, 25k and 100k membrane pellet fractions. Both IL-1β and the

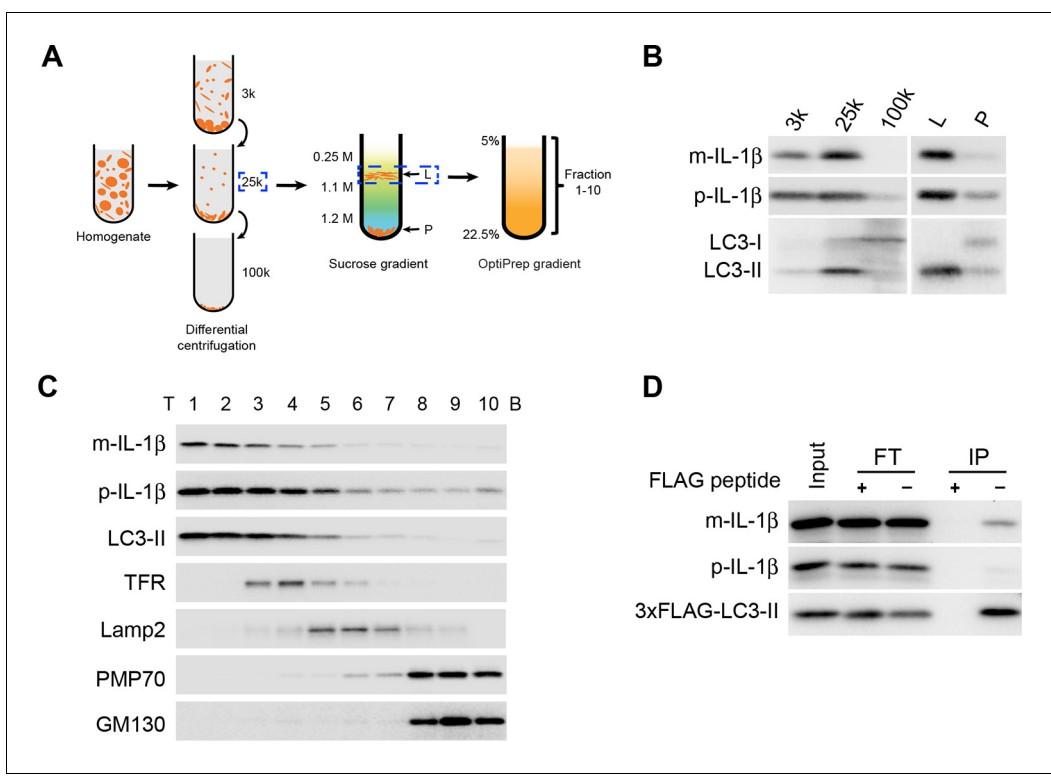

**Figure 2.** IL-1*β* vesicles co-fractionate with LC3 vesicles. (**A**) Membrane fractionation scheme. Briefly, HEK293T cells transfected with p-IL-1β and p-caspase-1 plasmids were starved in EBSS for 2 hr, collected and homogenized. Cell lysates were subjected to differential centrifugations at 3000×g (3k), 25,000×g (25k) and 100,000×g (100k). The level of IL-1β in each membrane fraction was determined by immunoblot. The 25k pellet, in which IL-1β was mainly enriched, was selected and a sucrose gradient ultracentrifugation was performed to separate membranes in the 25k pellet to the L (light) and P (pellet) fractions. The L fraction, which contained the majority of IL-1β, was further resolved on an OptiPrep gradient after which ten fractions from the top were collected. (**B,C**) Immunoblot was performed to examine the distribution of IL-1β, LC3 as well as the indicated membrane markers in the indicated membrane fractions. T, top; B, bottom (**D**) HEK293T cells transfected with p-IL-1β, p-caspase-1 and FLAG-tagged LC3-I plasmids were starved in EBSS for 2 hr. LC3 positive membranes were immunoisolated with anti-FLAG agarose from the 25 k pellet and the presence of IL-1β was determined by immunoblot analysis. FT, flowthrough

lipidated form of LC3 (LC3-II), a protein marker of autophagosome, were mainly enriched in the 25k membrane fraction (*Figure 2B*). We then separated the 25k membrane through a sucrose step gradient ultracentrifugation where both IL-1β and LC3-II co-distributed in the L fraction at the boundary between 0.25 M and 1.1 M layer of sucrose (*Figure 2B*). Further fractionation of the L fraction using an OptiPrep gradient showed co-fractionation of IL-1β with LC3-II (*Figure 2C*). To confirm the presence of IL-1β in the autophagosome, we performed immunoisolation of LC3-positive autophagosomes from the 25k fraction and found that IL-1β, especially the mature form, co-sedimented with autophagosomes (*Figure 2D*). Consistent with our observations, a recent study also showed a colocalization of IL-1β and LC3 in the form of puncta in macrophages (*Dupont et al., 2011*). These data demonstrate that at least a fraction of intracellular mature IL-1β associates with the autophagosome, possibly related to its role in IL-1β secretion.

To determine if IL-1β is localized to the phagophore in the absence of autophagosome completion, we fractionated membranes from ATG2-depleted cells, which are deficient in phagophore elongation and therefore fail to form mature autophagosomes (*Velikkakath et al., 2012*), and examined the distribution of LC3-II, which remains attached to immature phagophore membranes, and mature and precursor IL-1β. We performed the three-step fractionation described above. In control cells, IL-1β co-distributed with LC3-II in all three steps (*Figure 3*). Depletion of ATG2 did not affect the co-fractionation of IL-1β and LC3-II (*Figure 3*), indicating that IL-1β enters into the phagophore membrane before the completion of the autophagosome.

## Autophagosome formation is not required for entry of IL-1β into vesicles

We asked how IL-1β enters into the autophagosome. One possibility is engulfment through the closure of the phagophore membrane during autophagosome maturation as in the capture of autophagic cargo. In this scenario, closure of the phagophore to complete autophagosome formation would

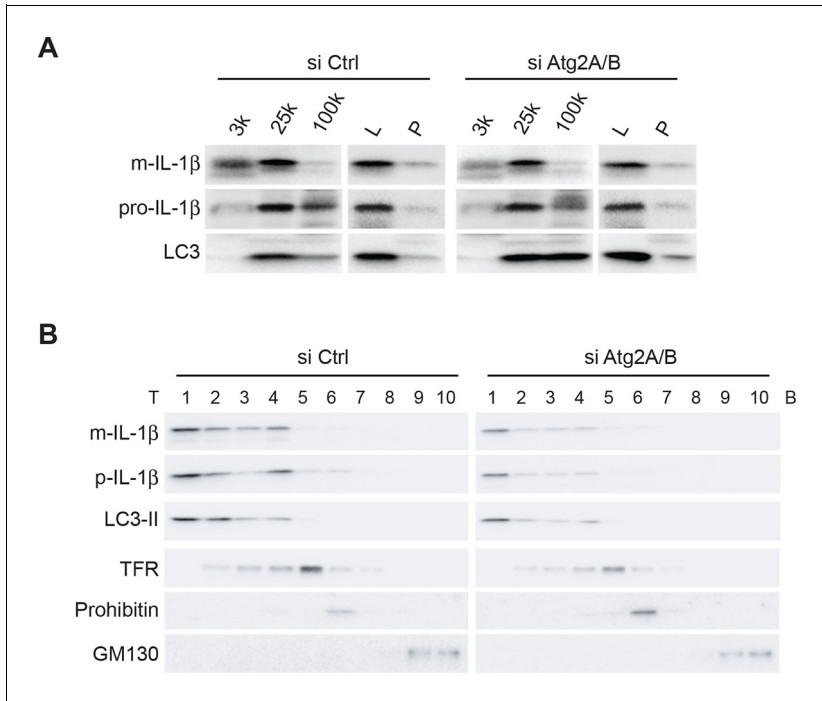

**Figure 3.** IL-1β co-distributes with LC3 in Atg2-depleted cells. (**A**) HEK293T cells were transfected with siRNAs against Atg2A and Atg2B followed with p-IL-1β and p-caspase-1 plasmids as shown in *Figure 1E*. The cells were starved in EBSS for 2 hr. Membrane fractions (3k, 25k, 100k (×g), L and P) were separated from the post-nuclear supernatant as depicted in *Figure 2B*. (**B**) Ten membrane fractions were collected from the OptiPrep gradient ultracentrifugation as depicted in *Figure 2C*. Immunoblot was performed to examine the distribution of IL-1β, LC3 as well as the indicated membrane markers. T, top; B, bottom.

be required to sequester IL-1β away from the cytoplasm. Alternatively, we considered the possibility that IL-1β may be translocated through a membrane into the lumen of the phagophore envelope and be sequestered from the cytoplasm even before the mature autophagosome is sealed. To test this possibility, we performed proteinase K protection experiments with the membranes from ATG2-depleted cells (*Figure 4A*). In control cells, p62 (an autophagic cargo) and a fraction of LC3-II (which was encapsulated after autophagosome completion), as well as mature IL-1β, were largely resistant to proteinase K digestion similar to the ER luminal protein, protein disulfide isomerase (PDI). In contrast, SEC22B, a membrane anchored SNARE protein exposed to the cytoplasm, was sensitive to proteinase K digestion (*Figure 4A*). Triton X-100 treatment permeabilized the membrane and rendered all proteins tested sensitive to proteinase K digestion (*Figure 4A*). This demonstrated that the majority of membrane localized IL-1β was sequestered within an organelle, likely the

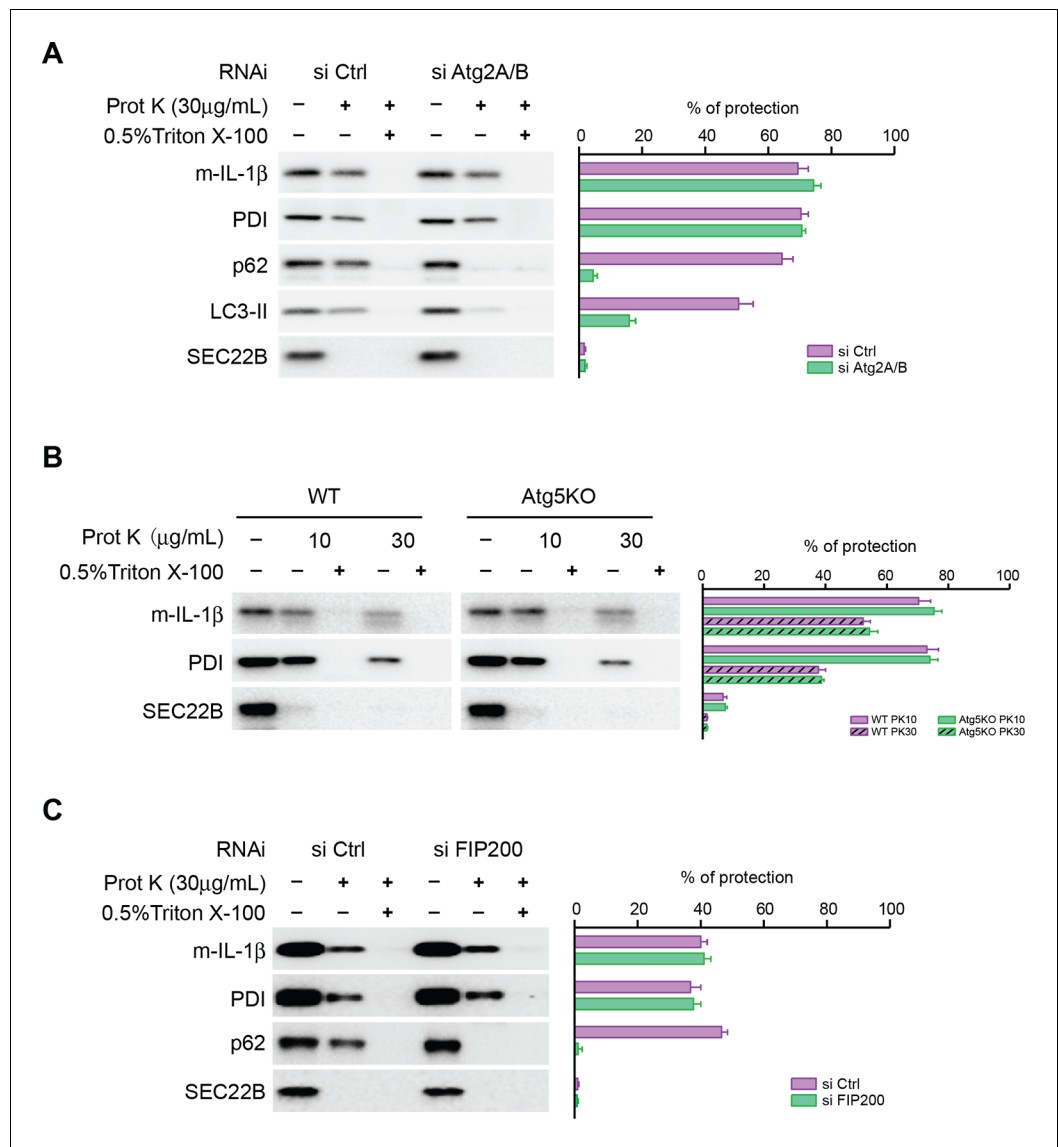

**Figure 4.** Closure of the autophagosome is not required for the entry of IL-1β into vesicles. (**A**) HEK293T cells were transfected with siRNAs against Atg2A and Atg2B followed by transfection with p-IL-1β and p-caspase-1 plasmids as shown in *Figure 1E*. The cells were starved in EBSS for 2 hr and proteinase K digestion was performed with the 25k membrane fractions. (**B**) Atg5 WT, KO MEFs were transfected with p-IL-1β and p-caspase-1 plasmids as shown in *Figure 1B*. The cells were starved in EBSS for 2 hr followed by proteinase K digestion as shown in (**A**). (**C**) HEK293T cells were transfected with siRNA against FIP200 followed by analysis of membrane entry of IL-1β as shown in (**A**). The level of proteinase K protection was calculated as the percentage of the total protein. Error bars represent standard deviations of at least three experiments.

autophagosome, as demonstrated by the fractionation results of *Figures 2 and 3*. However, the result did not pinpoint where within the autophagosome IL-1β was housed. In ATG2-depleted cells, p62 and LC3-II remained sensitive to proteinase K digestion, consistent with the hypothesis that ATG2 is essential for maturation and closure of the autophagosome (*Figure 4A*). However, in the same samples the majority of IL-1β resisted degradation by proteinase K treatment (*Figure 4A*), except on addition of Triton X-100 to permeabilize membranes. Although the precursor form of IL-1β remained associated with isolated autophagosome and phagophore membranes (*Figure 3*), the protein was degraded when membranes from normal and ATG2-depleted cells were treated with protease in the presence or absence of Triton X-100 (data not shown). Thus, the mature but not the precursor IL-1β appears to be transported into the phagophore.

A most recent study showed that small, closed double-membrane structures could be observed in ATG2-depleted cells (*Kishi-Itakura et al., 2014*). To rule out the possibility that IL-1β was engulfed by the small closed autophagosomes, we employed Atg5 KO MEFs in which the phagophore could not be closed (*Kishi-Itakura et al., 2014*; *Mizushima et al., 2001*). Similar to what we observed in ATG2-depleted cells, IL-1β was protected from proteinase K digestion in membranes from Atg5 KO MEFs (*Figure 4B*). In addition, IL-1β was sequestered within vesicles in FIP200 (another early factor in phagophore development (*Hara et al., 2008*)) knockdown cells (*Figure 4C*). These data indicate that the entry of IL-1β into the vesicle carrier is not dependent on the formation of the autophagosome. These results are inconsistent with a role for engulfment of IL-1β by the maturing phagophore and suggest instead that IL-1β may be translocated across a membrane into a vesicle precursor of the phagophore, possibly at a very early stage in the development of the organelle.

## Entry of IL-1β into the vesicle carrier requires protein conformational flexibility

We then sought to test if IL-1β could directly translocate across the membrane of a vesicle carrier. As protein unfolding is usually required for protein translocation, we adopted an approach used in many other circumstances wherein a targeted protein is fused to dihydrofolate reductase (DHFR), an enzyme whose three-dimensional structure is stabilized by the folate derivative aminopterin, hence providing a chemical ligand to impede the unfolding process (*Backhaus et al., 2004*; *Eilers and Schatz, 1986*; *Wienhues et al., 1991*). We first determined the secretion of the DHFR-fused IL-1β. As shown in *Figure 5A*, secretion of a mature IL-1β-DHFR fusion protein was enhanced by starvation similar to the untagged counterpart. Importantly, IL-1β-DHFR secretion was reduced in a dose-dependent manner in the presence of aminopterin (*Figure 5B*). Of notice, treatment of aminopterin did not completely abolish IL-1β secretion perhaps due to a cell death-induced release of IL-1β at high concentrations of aminopterin, as indicated by the release of a low level of tubulin into the medium fraction (*Figure 5B*). As a control, aminopterin did not reduce the secretion of untagged IL-1β, confirming its specific effect on DHFR (*Figure 5—figure supplement 1*). Fractionation of cells incubated with aminopterin showed a reduced level of IL-1β in the membrane fraction with a corresponding increase in the cytosol fraction (*Figure 5C*). The residual DHFR-tagged IL-1β associated with membranes from aminopterin-treated cells was sensitive to proteinase K digestion (*Figure 5D*), indicating that this pool of membrane-associated IL-1β did not translocate into the lumen of the vesicle. The data suggest that entry of IL-1β into a vesicle carrier involves a process of protein unfolding and translocation.

## IL-1β colocalizes with LC3 on the autophagosome envelope

If IL-1β is directly translocated across the membrane of a vesicle intermediate, fusion of these vesicles to form a double-membrane autophagosome would deposit IL-1β in the lumen between the two membranes of the autophagosome. To visualize the subcellular localization of IL-1β, we employed U2OS cells, which formed large and distinct autophagosomes after starvation. U2OS cells co-expressing p-IL-1β and p-caspase-1 secreted IL-1β in a starvation-enhanced and PI3K-dependent manner similar to HEK293T cells (*Figure 6—figure supplement 1*). To prepare for the subsequent fluorescence imaging, we also employed a FLAG-tagged m-IL-1β, which allowed us to directly determine the topological localization of the m-IL-1β. Secretion of m-IL-1β-FLAG from U2OS cells was stimulated by starvation and dependent on PI3K (*Figure 6—figure supplement 1*).

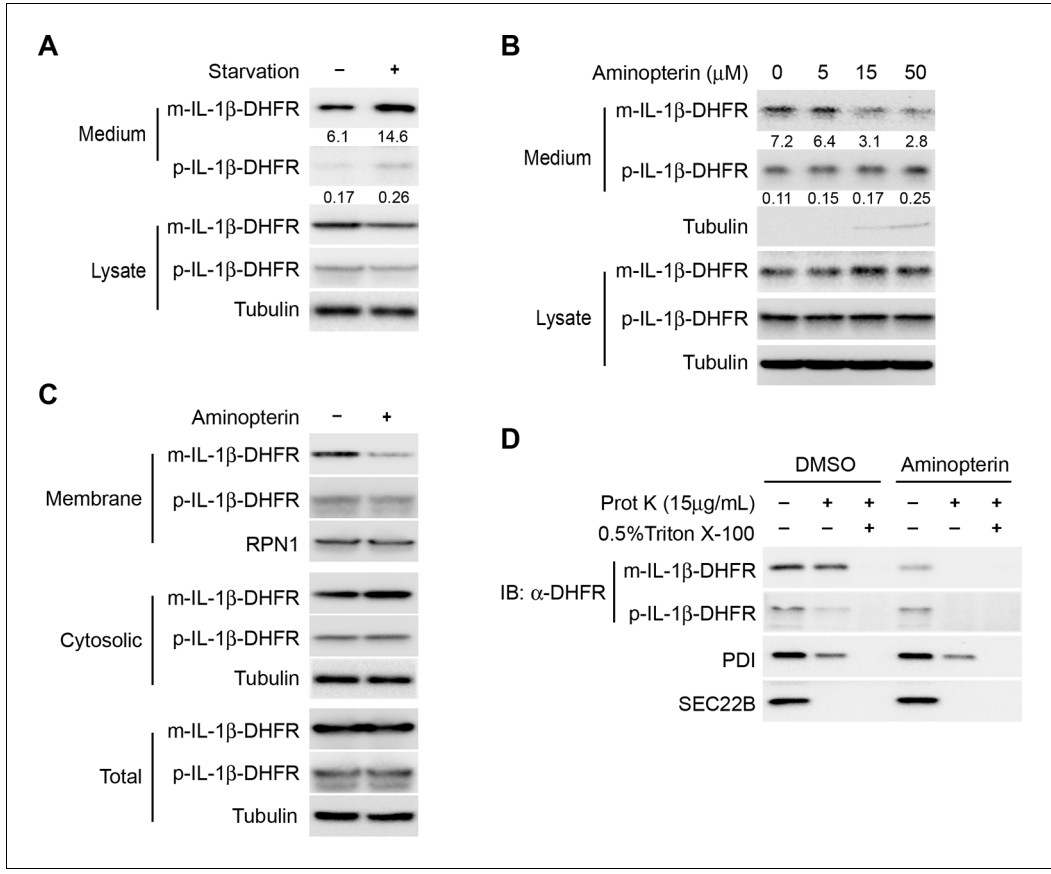

**Figure 5.** Protein unfolding is required for the entry of IL-1β into vesicles. (**A**) Secretion of DHFR-tagged IL-1β. HEK293T cells were transfected with p-IL-1β-DHFR and p-caspase-1 plasmids. After transfection (24 hr), the cells were treated with DMEM or EBSS for 2 hr. Release of IL-1β was determined as shown in *Figure 1*. (**B**) Secretion of IL-1β-DHFR was inhibited by aminopterin. HEK293T cells were transfected with p-IL-1β-DHFR and p-caspase-1 plasmids. After transfection (24 hr), the cells were treated with EBSS, or EBSS containing different concentrations of aminopterin as indicated for 15 min followed by determination of IL-1β secretion as shown in (A). Quantification of IL-1β secretion was calculated as the ratio between the amount of IL-1β in the medium and the total amount (the sum of IL-1β in both medium and lysate). (**C**) Less IL-1β enters into membrane in the presence of aminopterin. HEK293T cells were transfected with p-IL-1β-DHFR and p-caspase-1 plasmids. After transfection (24 hr), the cells were either untreated or treated with 5 μM aminopterin in EBSS for 2 hr. The membrane fraction was collected from the top fractions of a Nycodenz density gradient resolved from membranes in a 25k pellet as described in Material and Methods. The cytosolic fraction was collected as the supernatant after 100k×g centrifugation. All fractions were analyzed by immunoblotting using indicated antibodies. (**D**) IL-1β-DHFR is not protected from proteinase K in the presence of aminopterin. Nycodenz -floated membrane fraction collected as shown in (C) was subjected to proteinase K digestion and then analyzed by immunoblotting using indicated antibodies.

The following figure supplement is available for figure 5:

**Figure supplement 1.** Secretion of IL-1β is not affected by aminopterin.

To determine the topological distribution of IL-1β, we first performed confocal immunofluorescence labeling experiments. After starvation, cells were exposed to 40 μg/ml of digitonin to permeabilize the plasma membrane, harvested and washed with cold PBS to remove the excess cytosolic m-IL-1β-FLAG. In cells expressing either p-IL-1β and p-caspase-1, or m-IL-1β alone, LC3 and IL-1β were observed by confocal microscopy to localize together or adjacent to one another on the edge of ring-shaped autophagosomes (*Figure 6—figure supplement 2*). To further resolve these ring structures, we employed 3D STORM (*Huang et al., 2008*; *Rust et al., 2006*) super-resolution microscopy (*Hell, 2007*; *Huang et al., 2010*) (*Figure 6* and *Figure 6—figure supplements 3, 4* and

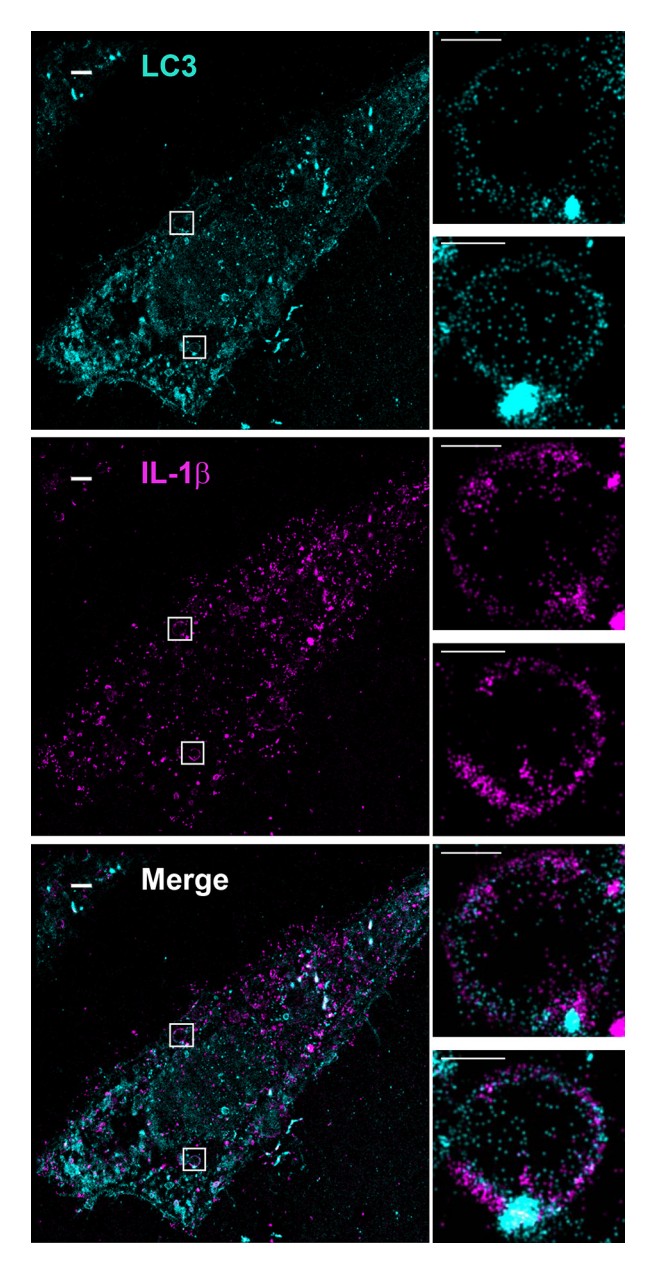

**Figure 6.** Topological localization of IL-1$\beta$ in the autophagosomal carrier determined by STORM. U2OS cells were transfected with a plasmid containing the expression cassette of FLAG-tagged mature IL-1β (m-IL-1β-FLAG). After transfection (24 hr), the cells were starved in EBSS for 1 hr followed by immunofluorescence labeling with mouse monoclonal anti-LC3 and rabbit polyclonal anti-FLAG antibodies. STORM analysis imaging and data analysis were performed as described in Materials and methods. Cyan, LC3; Magenta, IL-1β; Bars: 2 µm (original image) and 500 nm (magnified inset)

The following figure supplements are available for figure 6:

**Figure supplement 1.** Secretion of IL-1$\beta$ in U2OS cells.

**Figure supplement 2.** Localization of IL-1$\beta$ determined by confocal microscopy.

**Figure supplement 3.** Extra images for *Figure 6*.

**Figure supplement 4.** A minority of IL-1$\beta$ engulfed by autophagosome.

*Figure 6. continued on next page*

*Figure 6. Continued*

**Figure supplement 5.** Determination of the topological localization of IL-1β in the autophagosome and phagophore.

*Videos 1 and 2*). Ring-shaped autophagosomes positive for LC3 (cyan) formed after starvation. Some IL-1β (magenta) also organized in ring-shaped structures that co-localized with LC3 (*Figure 6* and *Figure 6—figure supplement 3*). Around 18 ring structures of IL-1β accounting for ~5% of the total IL-1β signal were observed in each cell. A 3D virtual Z-stack analysis confirmed the spatial co-distribution of LC3 and IL-1β on a ball-shaped vesicle (*Video 1 and 2*). The diameter of the structures double-labeled with LC3 and IL-1β are ~700 nm (larger structures up to 2 µm in diameter were also found) which is comparable to the size of the autophagosome. Occasionally, we also found IL-1β localized in the center of the ring structure, where cytoplasmic autophagic cargoes fill, surrounded by LC3 (*Figure 6—figure supplement 4*). This portion of IL-1β was possibly being engulfed by the autophagosome.

The visual detection of IL-1β localized to ring-shaped autophagosomes is consistent with our biochemical assays that place IL-1β in the intermembrane space between the outer and inner membrane of the autophagosome. We devised a further visual test of this conclusion using selective permeabilization of cell surface and intracellular membranes with digitonin and saponin, respectively (*Figure 6—figure supplement 5*). We compared antibody accessibility to IL-1β and DFCP1, a marker located on the cytosolic surface of the omegasome (a harbor for the phagophore) in both WT and Atg5 KO cells. Consistent with a cytosolic surface localization, DFCP1 was readily labeled in cells treated with digitonin alone (selectively permeabilizes the plasma membrane) in both WT and Atg5 KO cells (*Figure 6—figure supplement 5A–E*). In contrast, IL-1β was accessible to the antibody only after treatment with digitonin and saponin (gently permeabilizes the endomembrane) (*Figure 6—figure supplement 5A,B and F*) in WT cells. This by itself would not distinguish localization of IL-1β to the intermembrane space vs the cytoplasmic enclosed space of a mature autophagosome. However, in Atg5 KO cells where the phagophore precursor envelope remains open and exposed to the cytosol, saponin treatment was necessary to expose IL-1β to antibody and roughly half of the labeled structures coincided with the phagophore marker DFCP1 (*Figure 6—figure supplement 5C,D and F*). This visual assay further confirms the intermembrane localization of IL-1β in the phagophore and autophagosome.

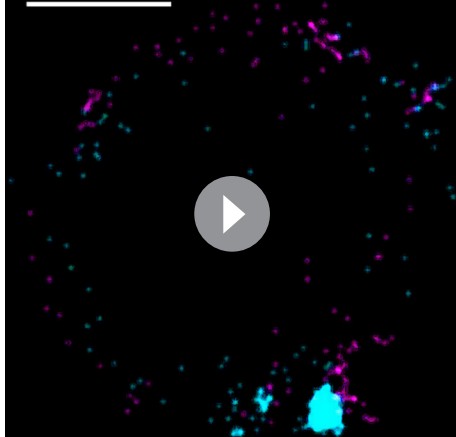

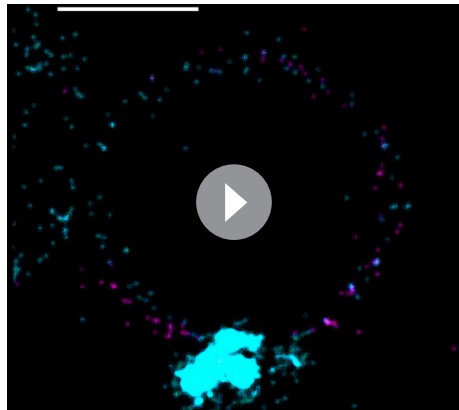

**Video 1.** 3D section of the magnified structure in *Figure 6* (upper one). The virtual Z-section thickness is 150 nm, and the step size is 50 nm. Cyan, LC3; Magenta, IL-1β; Bar 500 nm

**Video 2.** 3D section of the magnified structure in *Figure 6* (lower one). The virtual Z-section thickness is 150 nm, and the step size is 50 nm. Cyan, LC3; Magenta, IL-1β; Bar 500 nm

## Two KFERQ-like motifs are required for the entry of IL-1β into the vesicle carrier

In chaperone-mediated autophagy (CMA), cargoes are recognized by a KFERQ sequence motif for transport into the lysosome (*Dice et al., 1986*; *Kaushik and Cuervo, 2012*). We analyzed the primary sequence of IL-1β and found three KFERQ-like motifs on IL-1β including [127]LRDEQ[131], [132]QKSLV[136] and [198]QLESV[202] (*Figure 7A*). We mutated the glutamine, which has been shown to be essential for the function of the motif, as well as an adjacent amino acid in each motif (E130Q131, Q132K133 and Q198L199) to alanines and examined the secretion efficiency of these mutants. The

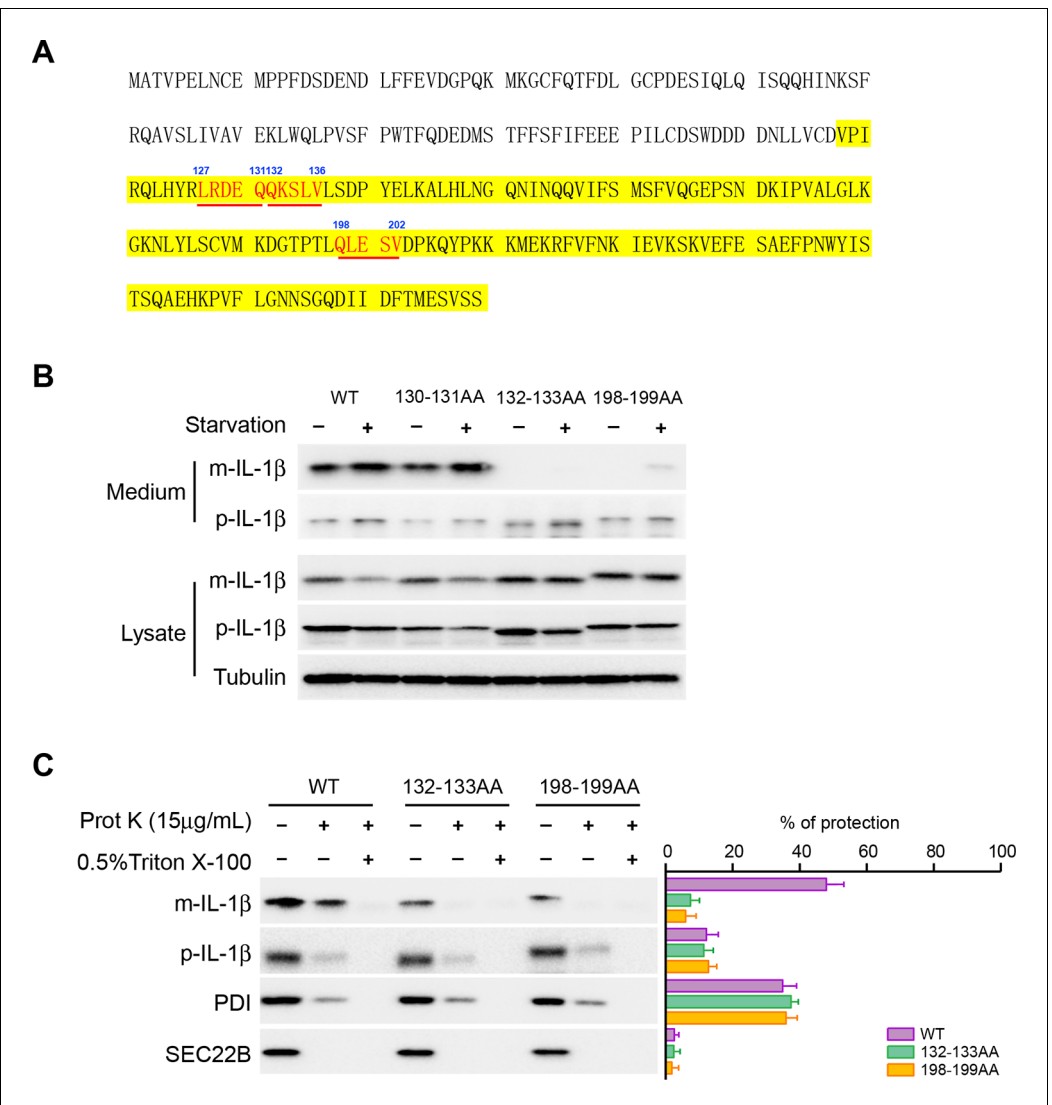

**Figure 7.** Mutation of the KFERQ-like motif affects IL-1β secretion and entry into vesicles. (A) Protein sequence of IL-1β. The yellow region indicates mature IL-1β. Three KFERQ-like motifs (aa127-131, aa132-136 and aa198-202) are highlighted in red underlined bold. (B) Secretion of IL-1β mutants. HEK293T cells were transfected with p-IL-1β-DHFR and p-caspase-1 plasmids. After transfection (24 hr), the cells were either treated with DMEM or EBSS for 2 hr. Secretion of IL-1β mutant proteins was detected by immunoblot. (C) IL-1β mutant 132-133AA or 198-199AA is accessible to proteinase K digestion. HEK293T cells were transfected with plasmids encoding p-caspase-1 and IL-1β mutant 132-133AA or 198-199AA. After transfection (24 hr), the cells were treated with EBSS for 2 hr. The 25k membrane fraction was collected and subjected to proteinase K digestion assay and then analyzed by immunoblot using indicated antibodies. The level of proteinase K protection was calculated as the percentage of the total protein. Error bars represent standard deviations of at least three experiments.

130-131AA mutant did not affect secretion of IL-1β (*Figure 7B*). However, the Q132K133 and Q198L199 mutations were both defective in secretion of mature IL-1β which instead accumulated in the cytoplasmic fraction (*Figure 7B*). A low level of release of the pro-forms persisted as seen with WT and mutant protein (*Figure 7B*). The cytoplasmic mature forms of the mutant proteins were less abundant in the membrane fraction compared with the WT mature IL-1β (*Figure 7C*, compare the lanes without proteinase K treatment). In addition, the membrane associated mutant IL-1β remained proteinase K accessible (less than 10% of protection compared with ~45% of WT IL-1β), demonstrating that these two KFERQ-like motifs are required for the membrane translocation of IL-1β (*Figure 7C*). Equal amounts of WT and mutant p-IL-1β associated with the membrane but both remained largely proteinase K accessible (*Figure 7C*).

## HSP90 is required for the entry of IL-1β into the vesicle intermediate

The chaperone proteins HSC70 and HSP90 have been reported to function in chaperone-mediated autophagy (CMA) (*Kaushik and Cuervo, 2012*; *Majeski and Dice, 2004*). HSP70 has also been implicated in autophagy and stress responses (*Murphy, 2013*). We performed shRNA-mediated knockdown of the three chaperone proteins to assess their potential role in the membrane translocation of IL-1β. Knockdown of Hsp90, but not of Hsp70 or Hsc70 substantially reduced IL-1β secretion (*Figure 8A*). As a control, knockdown of Hsc70 compromised CMA as indicated by the stabilization of a CMA cargo, GAPDH (*Figure 8—figure supplement 1A*). Moreover, secretion of mature IL-1β was inhibited in a dose-dependent manner by an HSP90 inhibitor geldanamycin (*Figure 8B*). In both experiments, mature IL-1β accumulated in the cytosol fraction at the expense of secretion. Knockdown of Hsp90 also rendered IL-1β accessible to proteinase K digestion (*Figure 8C*), consistent with a role for HSP90 in the translocation of IL-1β as opposed to some later secretion event. Furthermore, in a co-immunoprecipitation assay, HSP90 associated with m-IL-1β but not the translocation-deficient mutants Q132K133 and Q198L199 (*Figure 8D*). Although p-IL-1β also formed a complex with HSP90, the efficiency appeared lower than for m-IL-1β. These results suggest that HSP90 binds to a region of the mature IL-1β, including the essential residues Q132K133 and Q198L199, to promote the translocation event. Cleavage of p-IL-1β by caspase-1 may potentiate the recruitment of HSP90 to the mature form of IL-1β however chaperone binding is not required for this proteolytic event (*Figures 8D and 7B*).

In the CMA pathway, HSC70 and HSP90 play different roles. HSC70 binds to cargoes and delivers them into the lysosome as well as disassembling LAMP2A oligomers, whereas HSP90 is required for the oligomerization and stability of LAMP2A (*Bandyopadhyay et al., 2008*; *Chiang et al., 1989*). Co-immunoprecipitation indicated that IL-1β associates with HSP90 but not HSC70 (*Figure 8—figure supplement 1B*). In addition, knockdown of Lamp2A compromised CMA but did not affect the secretion of IL-1β, and disruption of the lysosome did not result in the release of IL-1β from the membrane carrier (*Figure 8—figure supplement 1C–E*). These data suggest that the translocation of IL-1β into the vesicle carrier is mechanistically distinct from CMA.

We next asked if starvation regulated the association between HSP90 and IL-1β. We performed an HSP90 co-immunoprecipitation experiment with cytosol prepared from cells grown in nutrient-rich or starvation conditions (*Figure 9A*). Starvation led to a ~2.5 fold increase of the association of HSP90 and IL-1β (*Figure 9A*). This increase was likely not due to starvation-stimulated processing of p-IL-1β because starvation had no effect on the cleavage of mutant forms of IL-1β unable to bind HSP90 (*Figure 7B*). Starvation led to a ~ 2 fold increase in the membrane localization and cytosolic depletion of mature IL-1β (*Figure 9B*). Starvation may stimulate the recruitment of a complex of m-IL-1β/HSP90 to the membrane responsible for IL-1β translocation (*Figure 9B*).

## Discussion

Genetic and cell biological studies have implicated autophagy in the transport of several leaderless cargoes to the extracellular space (*Bruns et al., 2011*; *Dupont et al., 2011*; *Duran et al., 2010*; *Manjithaya et al., 2010*). Unconventional secretory cargoes, such as IL-1β and Acb1, have been shown to have overlapping requirements with formation of the autophagosome or its precursor suggesting that the autophagosome may physically convey these cargo proteins to the cell surface. A key question is if and how these cargoes engage the autophagosome and how this structure exports soluble cargo molecules. In this study, we probed the organelle association and molecular

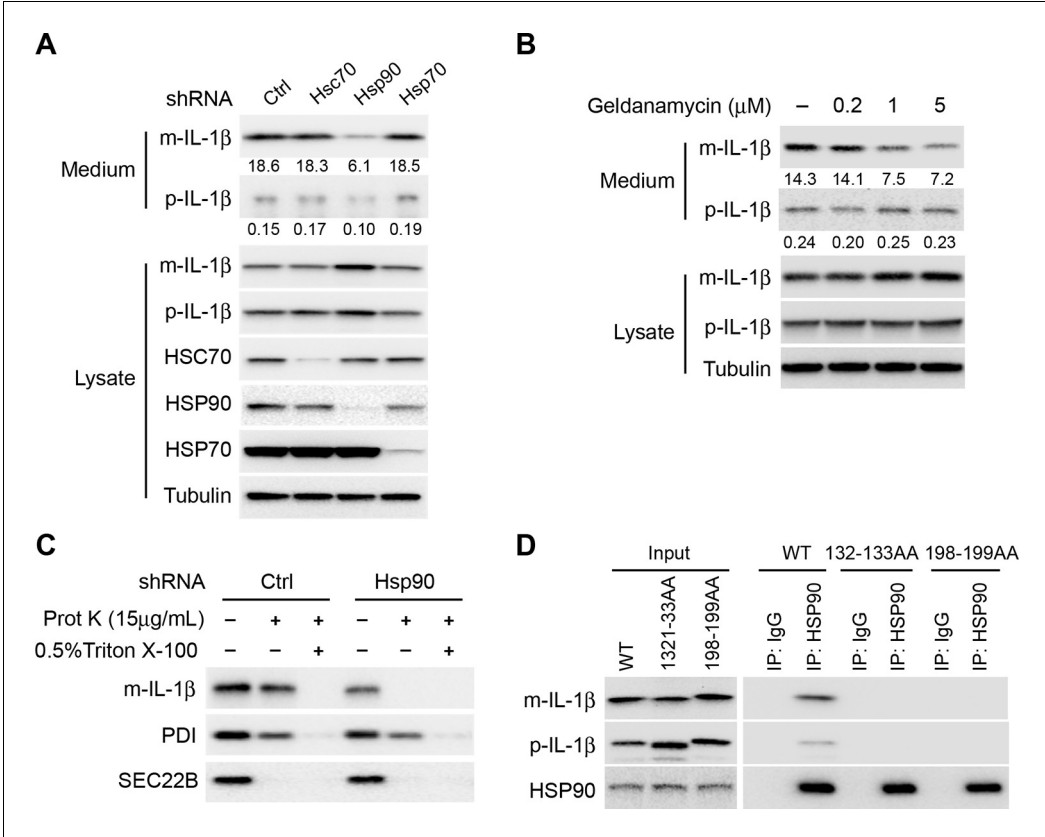

**Figure 8.** HSP90 is involved in the entry of IL-1β into vesicles. (**A**) Knockdown of Hsp90 inhibits IL-1β secretion. HEK293T cells were transduced with lentivirus carrying control (Ctrl) shRNA or shRNA against Hsc70, Hsp90 or Hsp70. Then the cells were transfected with p-IL-1β and p-caspase-1 plasmids. After transfection (24 hr), the cells were cultured in EBSS for 2 hr followed by determination of IL-1β secretion by immunoblot. (**B**) IL-1β secretion is reduced in the presence of HSP90 inhibitor geldanamycin. HEK293T cells were transfected with p-IL-1β and p-caspase-1 plasmids. After transfection (24 hr), the cells were treated with EBSS containing different concentrations of geldanamycin as indicated. Immunoblot was performed as shown in *Figure 1*. Quantification of IL-1β secretion was calculated as the ratio between the amount of IL-1β in the medium and the total amount (the sum of IL-1β in both medium and lysate). (**C**) IL-1β remains accessible to proteinase K in Hsp90 knockdown cells. HEK293T cells were transduced with lentivirus carrying control (Ctrl) shRNA or shRNA against Hsp90. Then the cells were transfected with p-IL-1β and p-caspase-1 plasmids. After transfection (24 hr), the cells were cultured in EBSS for 2 hr. The 25k membrane fraction was collected and digested with proteinase K and then analyzed by immunoblotting using indicated antibodies. (D) Association of HSP90 with IL-1β WT and mutants. HEK293T cells transfected with p-caspase-1 and IL-1β mutant 132-133AA or 198-199AA were starved in EBSS for 2 hr. Immunoprecipitation (IP) with anti-HSP90 antibody coupled to protein G-agarose was performed, followed by an immunoblot with anti-IL-1β and anti-HSP90 antibodies.

The following figure supplement is available for figure 8:

**Figure supplement 1.** Translocation of IL-1β is mechanistically different from CMA.

requirements for the secretion of one such unconventional cargo protein, IL-1β. Using surrogate cell lines rather than macrophages to reconstitute autophagy-mediated secretion of IL-1β (*Figure 1*), we find mature IL-1β localized to the lumen of the membrane in early intermediates and mature auto-phagosomes (*Figures 2–4, 6*). This surprising location may help to explain how mature IL-1β is secreted in a soluble form to the cell surface (*Figure 9C*). However, localization to the lumen between the two membranes of the autophagosome would require that IL-1β is translocated from the cytoplasm across the membrane precursor of a phagophore, rather than being engulfed as the phagophore membrane matures by closure into an autophagosome. Our evidence suggests that IL-

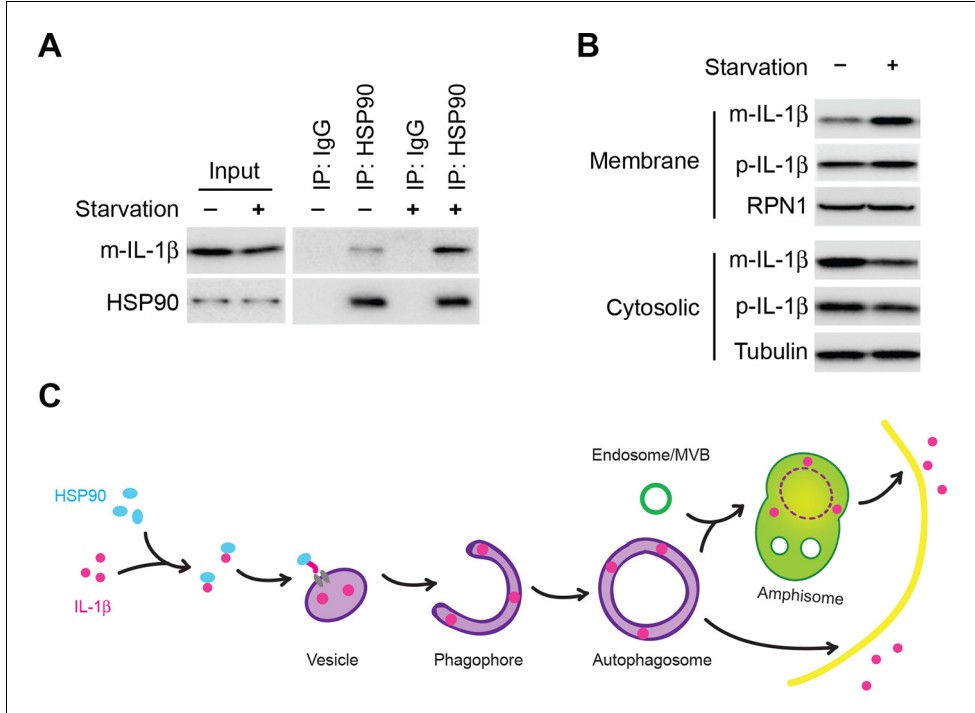

**Figure 9.** Induction of autophagy enhances the membrane incorporation of IL-1$\beta$. (**A**) Starvation enhances the association of IL-1β with HSP90. HEK293T cells transfected with p-IL-1β and p-caspase-1 were cultured in DMEM or EBSS for 2 hr. Immunoprecipitation with anti-HSP90 antibody was performed followed by an immunoblot with anti-IL-1β and anti-HSP90 antibodies. (**B**) Starvation promotes the entry of IL-1β into the membrane fraction. HEK293T cells transfected with p-IL-1β and p-caspase-1 were cultured in DMEM or EBSS for 2 hr. The membrane fraction was collected from the top fractions of a Nycodenz density gradient resolved from membranes in a 25k pellet as described in Material and Methods. The cytosolic fraction was collected as the supernatant after 100k×g centrifugation. Immunoblot was performed to determine the levels of IL-1β in both fractions. (**C**) A proposed model for autophagy-mediated IL-1β secretion. Cytosolic IL-1β associates with HSP90 which facilitates the translocation of IL-1β into the lumen of a vesicle carrier which later either turns into a phagophore and an autophagosome or fuses with them. IL-1β localizes between the outer and inner membrane after the double membrane autophagosome forms. The topological distribution ensures the secretion of IL-1β in a soluble form. The IL-1β-containing autophagosome may directly fuse with the plasma membrane or further fuse with a MVB followed by fusion with the plasma membrane.

1β must unfold or be held in an unfolded state to promote membrane translocation (*Figure 5*) and that a complex sorting signal in the mature portion of IL-1β interacts with HSP90 to deliver the chaperone and its cargo to a site on a phagophore precursor membrane where the mature species is translocated (*Figures 7–9*).

The unconventional secretory cargo fibroblast growth factor 2 (FGF2) has been shown to directly translocate across the plasma membrane as a folded protein without the apparent aid of chaperones (*Backhaus et al., 2004*; *Steringer et al., 2015*). Unlike FGF2, the entry of IL-1β into the autophagosomal carrier appears to be dependent on protein unfolding in a conformational state that may be fostered by the association of HSP90 with two KFERQ-like sequences within the mature portion of IL-1β (*Figure 5 and 8*). This translocation mechanism appears superficially similar to another delivery process termed HSC70-dependent CMA in which autophagic cargoes bearing KFERQ targeting motifs are directed into the lysosome for degradation. Indeed, using a cell-free approach to study the import of CMA cargo into isolated lysosomes, Salvador et al. (2000) reported that DHFR fused to a CMA cargo is blocked in translocation by addition of methotrexate, a drug that stabilizes DHFR to unfolding, just as we have shown that IL-1β fused to DHFR is blocked in cells treated with a cell permeable folate analog, aminopterin (*Wei et al., 2013*). In our fractionation study, IL-1β distributed in LC3-positive autophagosomal carriers that were separated from the lysosome marker LAMP2, the

proposed receptor or channel for uptake of CMA cargo (*Kaushik and Cuervo, 2012*) (*Figure 2B*). This observation, together with the involvement of a different chaperone i.e. HSP90, suggests distinct routes for IL-1β and cargoes of the CMA pathway.

The target membrane for IL-1β translocation may be a vesicle that could fuse with or form the autophagosome. We find that mature IL-1β can be detected within protease inaccessible membranes in cells blocked early in the autophagic pathway (e.g. ATG5 null cells and cells depleted of FIP200, both of which block at a stage prior to the lipidation of LC3). The identity of the vesicle carrier is unknown and could be any one of those reported to supply membrane to the formation of the autophagosome (*Ge et al., 2014a*; *Lamb et al., 2013*). Although we have ruled out the involvement of LAMP2A IL-1β translocation, it is likely that a membrane receptor locating on the membrane of the vesicle carrier, perhaps a functional equivalent of LAMP2A, recruits the protein complex of HSP90 and IL-1β, therefore designating the correct membrane targeting of IL-1β. In addition, a protein conducting channel may be involved in the translocation of IL-1β into the membrane. It seems unlikely that a standard translocation channel, such as SEC61, is involved in this import process, but no current evidence bears on this point.

The exact route by which the autophagosome delivers mature IL-1β to the cell surface as well as how it avoids fusion with degradative lysosome remains obscure, possibly involving interaction with the multi-vesicular body or some form of lysosome as a prelude to fusion at the cell surface (*Figure 9C*), and this process may require selective recruitment of membrane sorting and targeting factors such as Rabs and SNAREs. Fusion of the autophagosome directly with the plasma membrane would lead to the release of soluble IL-1β available to trigger an inflammatory response in the surrounding tissue. If mature IL-1β were engulfed within the cytoplasmic interior of the autophagosome, fusion of this organelle at the cell surface might release an intact vesicle corresponding to the inner membrane-enclosed cytoplasmic compartment of the autophagosome. We found mature IL-1β secreted by macrophages or in our surrogate cell system to be completely soluble, thus inconsistent with the engulfment model (data not shown). An alternative possibility may be that the autophagosome fuses with another intracellular organelle such as the MVB or the lysosome under conditions where the inner membrane of the autophagosome is degraded. If so, mature IL-1β would be available for secretion if the combined organelle (amphisome, *Figure 9C*) fused with the plasma membrane. However, for this model to be viable, the mature IL-1β released on dissolution of the autophagosome inner membrane would have to withstand proteolytic attack such as may be encountered in an amphisome or lysosome. Because mature IL-1β is clearly sensitive to proteolysis (*Figure 4*), thus any compartment engaged in presenting autophagosomal content to the cell surface must be depleted of proteases. The nature of the organelle that delivers autophagosome content to the plasma membrane may be probed by selective ablation of different Rab proteins, e.g. Rab11, Rab27 and Rab35, which appear to be required for fusion of the MVB with the cell surface (*Hsu et al., 2010*; *Ostrowski et al., 2010*; *Savina et al., 2002*), or Rab27a and Rab38, implicated in the fusion of lysosomes at the cell surface (*Blott and Griffiths, 2002*; *Hume et al., 2001*; *Jager et al., 2000*).

## Materials and methods

### Plasmids and siRNA oligos
The plasmid encoding p-IL-1β was kindly provided by Russell Vance lab (University of California, Berkeley). The plasmids encoding FLAG-tagged p-caspase-1 and ATG4B (C74A) were from Addgene (Cambridge, MA). The 3×FLAG-LC3 plasmid was generated by PCR insertion of a 3×FLAG peptide into MYC-LC3 (provided by the Zhong lab, UT Southwestern, Dallas). P-IL-1β mutants 130-131AA, 132-133AA and 198-199AA were generated by PCR-based site-directed mutagenesis. The p-IL-1β-DHFR plasmid was constructed by subcloning the DHFR peptide from MTS-GFP-DHFR (provided by the Nickel lab, Heidelberg, Germany) into pro-IL-1β. The plasmid encoding the FLAG-tagged m-IL-1β was constructed by deleting the sequence encoding AA2-117 of pro-IL-1β and inserting a DNA sequence encoding a single FLAG before the stop codon by PCR-based mutagenesis. The resulting protein is the m-IL-1β with a FLAG at the C-terminus.

Small interference siRNAs against Hrs or TSG101 were purchased from Qiagen (Valencia, CA). The target sequence against Hrs was CCGGAACGAGCCCAAGTACAA. The target sequence against

TSG101 was CAGTTTATCATTCAAGTGTAA. A pool of four siRNAs against Atg2A, Atg2B, FIP200, LAMP2, GRASP55 or GRASP65 was purchased from Qiagen (Valencia, CA; GeneSolution siRNAs), Dharmacon (Lafayette, CO; siGENOME SMART pool siRNAs) or Thermo Scientific (Rockford, IL; siGENOME SMART pool siRNAs).

## Cell culture and transfection

HEK293T and U2OS cells were grown in a tissue culture facility. Atg5 KO and WT MEFs were generously provided by Noboru Mizushima (University of Tokyo, Japan). Cells were grown at 37°C in 5% CO2 and maintained in Dulbecco's modified Eagle's medium (DMEM) supplemented with 10% FBS. For starvation, the cells were incubated in Earle's Balanced Salt Solution (EBSS) for the indicated time durations in the absence or presence of the drugs indicated in the manuscript. Transfection of DNA constructs into cells was performed using X-tremeGENE HP (Roche, Indianapolis, IN) according to the manufacture's protocols. SiRNA transfection was performed on HEK293T cells with lipofectamine RNAiMAX (Invitrogen, Carlsbad, CA) according to the manufacture's protocols.

## ShRNA constructs and lentiviral transduction

ShRNA constructs targeting Hsc70, Hsp90 and Hsp70 were inserted into pLKO.1-puro vector (Addgene). As previously described (*Hubbi et al., 2013*; *Zhong et al., 2011*; *Zuo et al., 2012*), the following sequences were used: Ctrl, CAACAAGATGAAGAGCACCAA; Hsc70-A, GCCCGATTTGAAGAACTGAAT; Hsc70-B, GCAACTGTTGAAGATGAGAAA; Hsp90, CCTGTGGA-TGAATACTGTATT; Hsp70, GGCCAACAAGATCACCATC.

For lentiviral transduction, plasmid pLKO.1 carrying Ctrl, Hsc70, Hsp90 or Hsp70 was transfected into HEK293T cells along with lentiviral packaging plasmids pMD2.G and psPAX2 (Addgene) using X-tremeGENE HP to produce lentiviral particles to infect HEK293T cells. Transduced cells were selected using 2 µg/ml Puromycin (Invitrogen).

## Reagents and antibodies

We obtained EBSS from Invitrogen (Grand Island, NY); 3-methyladenine (3-MA), wortmannin (Wtm), aminopterin, geldanamycin, proteinase K, 4-(2-Aminoethyl)-benzenesulfonyl fluoride hydrochloride (AEBSF), cycloheximide, anti-FLAG M2 agarose and 3×FLAG tag peptide from Sigma (St. Louis, MO); Protein G-Sepharose beads from GE Healthcare (Piscataway, NJ); glycyl-L-phenylalanine-2-naphthylamide (GPN) from Santa Cruz Biotechnology (Dallas, TX).

Mouse anti-GM130, anti-transferrin receptor (TFR), anti-PMP70, anti-FLAG, anti-PDI, anti-tubulin and rabbit anti-Prohibitin-1, anti-RPN1, anti-SEC22B, anti-LAMP2, anti-LC3 and anti-ERGIC53 antibodies were described before (*Ge et al., 2013*, *2014b*). We purchased goat anti-IL-1β antibody from R&D Systems (Minneapolis, MN); rabbit anti-IL-1β, anti-LAMP2A and mouse anti-HSC70 antibodies from Abcam (Cambridge, MA); rabbit anti-Caspase-1, rabbit anti-CD63, mouse anti-GAPDH and goat anti-GRASP65 antibodies from Santa Cruz (Dallas, TX); rabbit anti-ATG2A and anti-p62 antibodies from MBL (Woburn, MA); rabbit anti-FLAG and anti-ATG2B antibodies from Sigma (St. Louis, MO); mouse anti-FIP200 antibody from Millipore (Billerica, MA); mouse anti-Hrs antibody from Enzo Life Sciences (Lörrach, Germany); mouse anti-TSG101 antibody from GeneTex (San Antonio, TX); mouse anti-HSP90 antibody from EMD (Billerica, MA); mouse anti-HSP70 antibody from Enzo Life Sciences (Plymouth Meeting, PA); mouse anti-DHFR antibody from BD Biosciences Pharmingen (San Diego, CA); rabbit anti-GRASP55 antibody from ProteinTech Group (Chicago, IL).

## Reconstitution of IL-1β secretion in HEK293T or MEF cells

HEK293T or MEF cells were transfected with plasmids encoding different forms of p-IL-1β, p-caspase-1 as well as other plasmids as indicated in figure legends. After transfection (24 hr), cell culture media were replaced with DMEM, EBSS or EBSS containing indicated drugs for indicated time durations. Media were collected and concentrated (20-fold) by a 10 kD Amicon filter (Millipore, Billerica, MA). Cells were lysed in SDS-PAGE loading buffer and analyzed by immunoblot analysis.

For RNAi experiments, cells were transfected with indicated siRNAs. After 6 hr, a similar plasmid transfection indicated above was performed. After 60 hr, IL-1β secretion was determined.

## Differential centrifugation and membrane fractionation

The procedure is modified from our previous work (*Ge et al., 2013*). Cells (ten 10-cm dishes) were cultured to confluency, harvested and homogenized in a 2.7× cell pellet volume of B1 buffer (20 mM HEPES-KOH, pH 7.2, 400 mM Sucrose, 1 mM EDTA) plus a cocktail of protease and phosphatase inhibitors (Roche, Indianapolis, IN) and 0.3 mM DTT by passing through a 22 G needle until ~85% lysis analyzed by Trypan Blue staining. Homogenates were subjected to sequential differential centrifugation at 3,000×g (10 min), 25,000×g (20 min) and 100,000×g (30 min, TLA100.3 rotor, Beckman) to collect the membranes sedimented at each speed. The 25,000×g membrane pellet, which contained the highest level of IL-1β, was suspended in 0.75 ml 1.25 M sucrose buffer and overlaid with 0.5 ml 1.1 M and 0.5 ml 0.25 M sucrose buffer (Golgi isolation kit; Sigma). Centrifugation was performed at 120,000×g for 2 hr (TLS 55 rotor, Beckman), after which two fractions, one at the interface between 0.25 M and 1.1 M sucrose (L fraction) and the pellet on the bottom (P fraction), were separated. IL-1β protein levels of the two fractions were then tested and the L fraction was selected and suspended in 1 ml 19% OptiPrep for a step gradient containing 0.5 ml 22.5%, 1 ml 19% (sample), 0.9 ml 16%, 0.9 ml 12%, 1 ml 8%, 0.5 ml 5% and 0.2 ml 0% OptiPrep each. Each density of OptiPrep was prepared by diluting 50% OptiPrep (20 mM Tricine-KOH, pH 7.4, 42 mM sucrose and 1mM EDTA) with a buffer containing 20 mM Tricine-KOH, pH 7.4, 250 mM sucrose and 1mM EDTA. The OptiPrep gradient was centrifuged at 150,000×g for 3 hr (SW 55 Ti rotor, Beckman) and subsequently ten fractions, 0.5 ml each, were collected from the top. Fractions were diluted with B88 buffer (20 mM HEPES-KOH, pH 7.2, 250 mM sorbitol, 150 mM potassium acetate and 5 mM magnesium acetate) and membranes were collected by centrifugation at 100,000×g for 1 hr. Samples were normalized using a measured level of phosphatidylcholine (*Ge et al., 2013*) and subjected to SDS-PAGE followed by immunoblot analysis with the indicated antibodies.

## Membrane flotation assay

Cells (five 10-cm dishes) transfected with indicated plasmids were starved in EBSS for 2 hr and harvested as indicated above. Membranes from a 25,000×g membrane pellet were resuspended in 300 μl 60% (wt/vol) Nycodenz (Accurate Chemical, Westbury, NY) in B88 buffer and transferred to a Beckman tube (Polycarbonate, 11 × 34 mm). Aliquots were overlaid with 600 μl of 40% Nycodenz in B88 buffer and 100 μl B88 buffer, and then centrifuged for 2 hr at 100,000×g (TLS 55 rotor, Beckman). Ten fractions were collected from top to bottom and analyzed by immunoblot. For determining the level of IL-1β in the membrane fraction, top fractions were combined and diluted with B88 buffer and membranes were collected by centrifugation at 100,000×g for 40 min followed by immunoblot analysis.

## Immunoisolation

HEK293T cells (ten 10-cm dishes) transfected with p-IL-1β, p-caspase-1 and 3×FLAG-LC3 were starved in EBSS for 2 hr and harvested as indicated above. Membranes from a 25,000×g pellet were collected, resuspended in immunoisolation buffer containing 25 mM HEPES, pH 7.4, 140 mM potassium chloride, 5 mM sodium chloride, 2.5 mM magnesium acetate, 50 mM sucrose and 2 mM EGTA. Anti-FLAG M2 agarose was added to a 1 ml membrane suspension with or without 0.2 mg/ml 3×FLAG tag blocking peptides and mixed by rotation at 4°C overnight. Beads with the associated membranes were washed with 1 ml immunoisolation buffer three times and membranes bound to the beads were eluted by incubating with 0.5 mg/ml of 3×FLAG peptides for 0.5 hr at room temperature. The eluted membranes were collected by centrifuging at 100,000×g for 40 min and analyzed by immunoblot.

## Proteinase K protection assay

The 25,000×g membrane pellet separated from cell homogenates were aliquoted into several fractions and resuspended in 30 μl of B88 or B88 containing indicated concentrations of proteinase K with or without 0.5% Triton X-100, and stored on ice for 30 min. The reactions were stopped by sequentially adding AEBSF and 3×SDS loading buffer, which were then heated in boiling water for 10 min and analyzed by immunoblot.

## Co-immunoprecipitation

Co-immunoprecipitation was performed as previously described (*Zhang et al., 2010*). Briefly, 24 hr after transfection, the cells were lysed on ice for 30 min in lysis buffer (50 mM Tris/HCl pH 7.4, 150 mM NaCl, 1 mM EDTA, 1% Triton X-100, 10% glycerol) with protease inhibitor mixture, and the lysates were cleared by centrifugation. The resulting supernatants were incubated with mouse anti-HSP90 or a mouse IgG control antibody overnight at 4°C and then precipitated with Protein G-Sepharose beads for 2 hr at 4°C. After washing, 3×SDS loading buffer was added to the beads and immunoblot was performed.

## Immunofluorescence labelling and STORM imaging

U2OS cells were starved in EBSS for 1 hr and permeabilized with 40 µg/ml of digitonin diluted in PBS on ice for 5 min. The cells were then washed once with cold PBS and immediately incubated with 4% cold paraformaldehyde for 20 min at room temperature. The cells were further permeabilized with 0.1% saponin diluted in PBS at room temperature for 10 min followed by blocking with 10% FBS diluted with PBS for 1 hr and primary antibody incubation for 1 hr. For confocal microscopy, procedures were as described previously (*Ge et al., 2014b*). For STORM, cells were washed three times with 0.2% BSA in PBS, followed by incubation with CF568 anti-mouse (Biotium) and Alexa Fluor 647 anti-rabbit (Invitrogen) secondary antibodies in 3% BSA in PBS for 1 hr at room temperature. Cells were washed three times before mounting for STORM imaging.

Dye-labeled cell samples were mounted on glass slides with a standard STORM imaging buffer consisting of 5% (w/v) glucose, 100 mM cysteamine, 0.8 mg/mL glucose oxidase, and 40 µg/mL catalase in 1M Tris-HCI (pH 7.5) (*Huang et al., 2008*; *Rust et al., 2006*). Coverslips were sealed using Cytoseal 60. STORM imaging was performed on a homebuilt setup based on a modified Nikon Eclipse Ti-U inverted fluorescence microscope using a Nikon CFI Plan Apo λ 100x oil immersion objective (NA 1.45). Dye molecules were photoswitched to the dark state and imaged using either 647- or 560-nm lasers (MPB Communications); these lasers were passed through an acousto-optic tunable filter and introduced through an optical fiber into the back focal plane of the microscope and onto the sample at intensities of ~2 kW cm$^{-2}$. A translation stage was used to shift the laser beams towards the edge of the objective so that light reached the sample at incident angles slightly smaller than the critical angle of the glass-water interface. A 405-nm laser was used concurrently with either the 647- or 560-nm lasers to reactivate fluorophores into the emitting state. The power of the 405-nm laser (typical range 0–1 W cm$^{-2}$) was adjusted during image acquisition so that at any given instant, only a small, optically resolvable fraction of the fluorophores in the sample were in the emitting state. Emission was recorded with an Andor iXon Ultra 897 EM-CCD camera at a framerate of 110 Hz, for a total of ~80,000 frames per image. Multicolor imaging was performed by imaging each color channel separately and sequentially using separate emission filters and excitation lasers. For 3D STORM imaging, a cylindrical lens was inserted into the imaging path so that images of single molecules were elongated in opposite directions for molecules on the proximal and distal sides of the focal plane (*Huang et al., 2008*). The raw STORM data was analyzed according to previously described methods (*Huang et al., 2008*; *Rust et al., 2006*).

## Acknowledgements

We thank Suzanne Pfeffer (Stanford University), Russell Vance, Jeremy Thorner, James Hurley, Roberto Zoncu (UC Berkeley) and Li Yu (Tsinghua University, China) for helpful information and advice on the study; Russell Vance (UC Berkeley), Walter Nickel (Heidelberg University, Germany), Noboru Mizushima (Tokyo University, Japan), Qing Zhong (UT Southwestern) and David King (UC Berkeley) for reagents; Bob Lesch and Ann Fisher cell culture facility (UC Berkeley) for technical assistance; MZ is an HHMI Associate. SK is a graduate student and is supported by College of Chemistry at UC Berkeley. LG was supported by a fellowship from the Jane Coffin Childs Fund (JCCF) and now supported by NIH Pathway to Independence Award (Parent K99/R00) National Institute of General Medical Sciences (Grant Number: 1K99GM114397-01). KX is a Pew Biomedical Scholar and is supported by College of Chemistry at UC Berkeley. RS is an Investigator of the HHMI and a Senior Fellow of the UC Berkeley Miller Institute.

# Additional information

## Competing interests

RS: Editor-in-Chief, *eLife.* The other authors declare that no competing interests exist.

## Funding

| Funder | Grant reference number | Author |
|---|---|---|
| Pew Charitable Trusts | | Samuel J Kenny<br>Ke Xu |
| National Institute of General Medical Sciences | 1K99GM114397-01 | Liang Ge |
| Jane Coffin Childs Memorial Fund for Medical Research | JCCF Postdoctoral Fellowship | Liang Ge |
| UC Berkeley College of Chemistry | | Ke Xu |
| Howard Hughes Medical Institute | | Randy Schekman |
| Adolph C. and Mary Sprague Miller Institute for Basic Research in Science, University of California Berkeley | | Randy Schekman |

The funders had no role in study design, data collection and interpretation, or the decision to submit the work for publication.

## Author contributions

MZ, Conception and design, Acquisition of data, Analysis and interpretation of data, Drafting or revising the article, Contributed unpublished essential data or reagents; SJK, Acquisition of data, Analysis and interpretation of data, Drafting or revising the article; LG, Conception and design, Acquisition of data, Analysis and interpretation of data, Drafting or revising the article; KX, RS, Conception and design, Analysis and interpretation of data, Drafting or revising the article

## Author ORCIDs

Liang Ge, http://orcid.org/0000-0002-7371-2039
Ke Xu, http://orcid.org/0000-0002-2788-194X
Randy Schekman, http://orcid.org/0000-0001-8615-6409

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
