## [Decision Letter]

Thank you for submitting your work entitled "Translocation of interleukin-1β into a vesicle intermediate in autophagy-mediated secretion" for peer review at *eLife*. Your submission has in principle been favorably evaluated by Tadatsugu Taniguchi (Senior editor) and three reviewers, one of whom, Noboru Mizushima, is a member of our Board of Reviewing Editors.

The reviewers have discussed the reviews with one another and the Reviewing editor has drafted this decision to help you prepare a revised submission; please note that there are some critical concerns that you will need to address to satisfy the reviewers.

Summary:

This paper describes a novel mechanism of unconventional secretion, in which IL-1β is first transported into the lumen of the phagophore (i.e., the space between the autophagosomal inner and outer membranes) and then secreted out by an unknown mechanism involving GRASP and some MVB proteins. This translocation depends on two KFERQ-like motifs, typically required for protein translocation across the lysosomal membrane via chaperone-mediated autophagy (CMA), and is also dependent of Hsp90, but not Hsc70. Although how IL-1β is secreted rather than being delivered to lysosomes and how GRASP and MVB proteins actually work in IL-1β secretion remains unclear, the present study focuses on the sequestration step. However, there are several major concerns outlined below.

Essential revisions:

1) The site of membrane translocation of IL-1β is important. The authors show that m-IL-1β is protease-protective in several autophagy mutants. The data of Atg2 KD and Atg5 KO could be reasonable because some intermediate structures can be generated in these cells. In contrast, phagophores are not generated in FIP200 KO cells. However, a significant amount of m-IL-1β is still resistant to proteinase K treatment in FIP200 cells. It suggests that IL-1β is sequestered in some membranous structures other than phagophores, which exist in a large amount comparable to that of phagophores in control cells. Although it is possible that these may be phagophore precursors, what is the assurance that this is phagophore-related?

An alternative mechanism, which should be ruled out, is that IL-1β is a CMA/MVB substrate that is first transported into the lysosomal/endosomal membrane and then injected into the inter-membrane space of the autophagosome upon fusion. In Figure 2, there are multiple fractions that contain LC3 and m-IL-1β. It is hard to draw conclusions from this about localization of IL-1β in autophagosome. LC3 is also known to be associated with other membranes, particularly a transfected LC3. In Figure 2, there is some p- and m-IL-1β associated with lysosomes (Lamp2). How can they eliminate the possibility that some m-IL-1β might be generated by CMA? CMA depends on KFERQ-like sequences, as well as Hsc70 and Hsp90. Here, m-IL-1β production was dependent on Hsp90 depletion using shRNA or inhibition using a drug, but apparently not by Hsc70 depletion. There is an experiment (Figure 8) using shRNA to hsc70, but this experiment needs a control for CMA itself making it hard to judge whether CMA was fully inhibited, but m-IL-1β production was not. Otherwise, it cannot be concluded that CMA is not involved in m-IL-1β production.

2) Although the present study includes many biochemical data, the morphological evidence showing that IL-1β is indeed present in the inter-membrane space is very limited. The fluorescence microscopy data in Figure 6 are suggestive, but not conclusive. It is unclear how representative these ring-shaped structures are. As this is the main argument of this study, it is recommended to try immune-electron microscopy to detect IL-1β in the inter-membrane space of the autophagosome.

Minor points:

1) Figure 2: It is not clear what "enriched" means as there are no controls for the amount of proteins. For example, it is not clear how much protein were in the 3K fractions.

2) In Figure 3, it appears that m-IL-1β is mostly in the 3K fraction, particularly at the siControl panel, in contrast to what shown in Figure 2. This raises concern about the validity of this technique. These data need to be more carefully described.

3) Figure 4: The assertion that p62, LC3 and IL-1β were resistant to proteinase K digestion is not correct. They are all partially degraded by proteinase K, albeit less than SEC22B.

4) Figure 7: The expression of the mutants seem to be much less than WT, which could have been a confounding factor in the interpretation of the data. How much of what seen in Figure 7 lack of secretion vs lack of detection due to low expression of the mutants, although they seem to be expressed well in Figure 7 but not in Figure 7 raising questions about reproducibility.

5) Figure 9: This study is unable to distinguish whether it is transported across the membrane of a vesicular compartment distinct from the phagophore or not. Figure 9 should be altered to reflect this.

6) Figure 9: What would divert the autophagosome with m-IL-1β in the intermembrane space from fusing with lysosomes, which would dump the IL-1β into the lumen of lysosomes? This may be discussed.

7) Some references such as Liu et al., 2014 and Shirasaki et al., 2014 are mentioned in the text but not included in References. Please check them carefully.

---

## [Author Response]

Essential revisions:1) The site of membrane translocation of IL-1β is important. The authors show that m-IL-1β is protease-protective in several autophagy mutants. The data of Atg2 KD and Atg5 KO could be reasonable because some intermediate structures can be generated in these cells. In contrast, phagophores are not generated in FIP200 KO cells. However, a significant amount of m-IL-1β is still resistant to proteinase K treatment in FIP200 cells. It suggests that IL-1β is sequestered in some membranous structures other than phagophores, which exist in a large amount comparable to that of phagophores in control cells. Although it is possible that these may be phagophore precursors, what is the assurance that this is phagophore-related?

The reviewers raise a valid concern. Our data demonstrates a relationship of secretion of IL-1β to autophagy and the transition of IL-1β through an autophagosome carrier by showing that 1) the secretion of IL-1β is compromised in the absence of several essential autophagy genes (Figure 1); 2) IL-1β resides in LC3-positive compartments during secretion as demonstrated by fractionation, immunoisolation and imaging approaches (Figure 2, Figure 3, Figure 6 and Figure 6—figure supplement 2–Figure 6—figure supplement 4); and 3) IL-1β may also localize to phagophore structures before the completion of double-membrane autophagosome as shown by the co-fractionation of IL-1β with LC3 in Atg2 RNAi cells where the LC3-positive vesicles are still in the state of phagophore or small closed double-membrane structures. To further confirm the relationship of IL-1β with phagophore, we have now performed immunofluorescence experiments by analyzing the localization of IL-1β and DFCP1, an omegasome marker that decorates the phagophore, in WT U2OS cells and Atg5 KO MEFs. In both cell lines, more than half of IL-1β colocalized with DFCP1 confirming a relationship of IL-1β with the phagophore (Figure 6—figure supplement 5). Even though we demonstrate a transition of IL-1β through phagophore and autophagosome during secretion, it is possible that the site of IL-1β translocation may be neither one of them, as a deficiency of FIP200 did not compromise the entry of IL-1β into the vesicles (Figure 4). As indicated in the Discussion of our manuscript, the site of IL-1β translocation could be a vesicle that could become a phagophore, especially in the absence of FIP200. However, we could not completely rule out the possibility that the site of IL-1β translocation is a combination of phagophore, autophagosome and a vesicle other than in WT cells. We agree with the reviewers that to pinpoint the site of membrane translocation of IL-1β is important. However, to completely define the targeting membrane for IL-1β translocation, it is important to identify the translocon, or an equivalent, directly responsible for IL-1β translocation, as well as to reconstitute the translocation process in vitro. This will take at least 1-2 years’ effort. Therefore, we think that these objectives should be our future work and beyond the scope of the current manuscript. We hope that the reviewers agree with our opinion. We have further discussed these points in the Discussion section.

An alternative mechanism, which should be ruled out, is that IL-1β is a CMA/MVB substrate that is first transported into the lysosomal/endosomal membrane and then injected into the inter-membrane space of the autophagosome upon fusion. In Figure 2, there are multiple fractions that contain LC3 and m-IL-1β. It is hard to draw conclusions from this about localization of IL-1β in autophagosome. LC3 is also known to be associated with other membranes, particularly a transfected LC3. In Figure 2, there is some p- and m-IL-1β associated with lysosomes (Lamp2). How can they eliminate the possibility that some m-IL-1β might be generated by CMA? CMA depends on KFERQ-like sequences, as well as Hsc70 and Hsp90. Here, m-IL-1β production was dependent on Hsp90 depletion using shRNA or inhibition using a drug, but apparently not by Hsc70 depletion. There is an experiment (Figure 8) using shRNA to hsc70, but this experiment needs a control for CMA itself making it hard to judge whether CMA was fully inhibited, but m-IL-1β production was not. Otherwise, it cannot be concluded that CMA is not involved in m-IL-1β production.

This is a reasonable concern that we have now addressed. To rule out CMA, we analyzed CMA in the absence and presence of the shRNA against Hsc70 (Figure 8—figure supplement 1). GAPDH, a CMA substrate, was decreased 2-fold after serum starvation and inhibition of protein synthesis, and this degradation was compromised by knockdown of Hsc70 (Figure 8—figure supplement 1). However, in this condition, the secretion and membrane entry of IL-1β were not affected indicating that this key chaperone for CMA is not involved in the secretion of IL-1β (Figure 8). To further rule out the possibility of CMA, we knocked down Lamp2A, the lysosomal receptor required for CMA, which again blocked the reduction in CMA-induced degradation of GAPDH but did not affect the secretion of IL-1β (Figure 8—figure supplement 1). In addition, we did co-immunoprecipitation experiments and found that IL-1β associates with HSP90 but not HSC70 (Figure 8—figure supplement 1). HSC70 and HSP90 plays different roles in CMA, HSC70 binds CMA substrates and is involved to deliver substrates to the lysosome. HSP90 is required for LAMP2A oligomerization and stability on the lysosomal membrane, which is essential for substrate translocation. However, in the case of IL-1β secretion, HSP90 acts as a functional equivalent of HSC70 for CMA. We also used glycyl-L-phenylalanine-2-naphthylamide (GPN), a lysosome-disrupting cathepsin C substrate, to disrupt the lysosome, which led to a dramatic decrease of cathepsin D (Figure 8—figure supplement 1), a lysosome luminal hydrolase, in the membrane fraction. However, the level of IL-1β in the membrane fraction was not affected by lysosome disruption (Figure 8—figure supplement 1) indicating the majority of IL-1β is not in the lysosome, or at least the digestive lysosome, during secretion. In summary, all those data indicate that the translocation of IL-1β is mechanistically different from CMA.

It is unlikely that IL-1β is taken up by the MVB because 1) it should lead to the secretion of IL-1β in the form of exosomes but IL-1β is secreted from cells as a soluble protein; 2) it will not result in the topological distribution of IL-1β in the intermembrane of the autophagosome; 3) enclosure within an intralumenal vesicle should not depend on membrane translocation. However, we could not rule out the possibility that IL-1β could be directly translocated into an MVB. To examine the exact membrane target for translocation of IL-1β, our plan is to devise a cell-free reaction that reproduces this step, and this clearly extends beyond the scope of this manuscript.

2) Although the present study includes many biochemical data, the morphological evidence showing that IL-1β is indeed present in the inter-membrane space is very limited. The fluorescence microscopy data in Figure 6 are suggestive, but not conclusive. It is unclear how representative these ring-shaped structures are. As this is the main argument of this study, it is recommended to try immune-electron microscopy to detect IL-1β in the inter-membrane space of the autophagosome.

We agree that immune-EM would further strengthen our conclusion. However, such experiments would require extensive preparatory work and involve a double-immunogold labeling protocol that we currently do not have access to in our lab. As a compromise, we performed selective permeabilization of cell membranes by two detergents, digitonin and saponin (Figure 6—figure supplement 5). The former one selectively permeabilizes plasma membrane at low temperature and the later one mildly permeabilizes the endomembrane. As shown in Figure 6—figure supplement 5, after digitonin treatment, DFCP1 could be labeled by antibodies, as it decorates the cytosolic surface of omegasome (Figure 6—figure supplement 5, green and blue). Interestingly, IL-1β puncta could not be labeled after digitonin treatment, but they appeared after saponin treatment in WT cells (Figure 6—figure supplement 5, red). As two of the three possibilities of IL-1β localization illustrated in Figure 6—figure supplement 5 panel A, IL-1β puncta should be protected by membrane (either on the inner membrane surface or intermembrane space according to the ring structure obtained from STORM). To discern the two possibilities, we performed the experiment in Atg5 KO cells where the phagophore is not closed. Similarly, the IL-1β puncta could be labeled only after saponin treatment (Figure 6—figure supplement 5, red). According to the illustration in Figure 6—figure supplement 5 panel C, the probability is that IL-1β localizes to the intermembrane space. In addition to these experiments, we also quantified the number of ring structures of IL-1β obtained from the STORM. On average, there are ~18 ring structures in each cell that consists of ~5% of the total IL-1β signal. These ring structures appear representative of the localization of IL-1β in the autophagosome as we have found many of these structures in different cells from independent experiments.

Minor points:1) Figure 2: It is not clear what "enriched" means as there are no controls for the amount of proteins. For example, it is not clear how much protein were in the 3K fractions.

This is a valid point as we neglected to state how the sample sizes were normalized to membrane. For this we employed our previous protocols on membrane fractionation. Each membrane fraction was normalized with an abundant phospholipid, phosphatidylcholine. We have added the explanations in the Materials and methods and within the figure legend.

2) In Figure 3, it appears that m-IL-1β is mostly in the 3K fraction, particularly at the siControl panel, in contrast to what shown in Figure 2. This raises concern about the validity of this technique. These data need to be more carefully described.

We thank the reviewers for pointing out this concern. Indeed, the results appear inconsistent with our other repetitions of this experiment therefore we have reprobed the IL-1β samples for Figure 3 and present the new results on the same samples as were used for the rest of this experiment in Figure 3.

3) Figure 4: The assertion that p62, LC3 and IL-1β were resistant to proteinase K digestion is not correct. They are all partially degraded by proteinase K, albeit less than SEC22B.

We thank the reviewers for the suggestion. We have changed the description in the new manuscript.

4) Figure 7: The expression of the mutants seem to be much less than WT, which could have been a confounding factor in the interpretation of the data. How much of what seen in Figure 7 lack of secretion vs lack of detection due to low expression of the mutants, although they seem to be expressed well in Figure 7 but not in Figure 7 raising questions about reproducibility.

The expression level of WT and mutants are equal. As indicated in the manuscript, there is less mutant protein associated with the membrane fraction because of reduced translocation. In Figure 7, the mutants are lower because we used the membrane fraction for the proteinase K protection assay. To address the concern of unequal level in the membrane, we quantified the percentage of proteinase K protection. As shown in Figure 7, ~45% of WT IL-1β was protected from digestion compared with less than 10% of the mutants.

5) Figure 9: This study is unable to distinguish whether it is transported across the membrane of a vesicular compartment distinct from the phagophore or not. Figure 9 should be altered to reflect this.

We did not highlight the idea of phagophore in Figure 9 in the initial version of the manuscript as the structure we drew is not cup-shaped. We have added more explanations in this Figure and the figure legend.

6) Figure 9: What would divert the autophagosome with m-IL-1β in the intermembrane space from fusing with lysosomes, which would dump the IL-1β into the lumen of lysosomes? This may be discussed.

We thank the reviewers for the suggestion. We have discussed this point in the Discussion section.

7) Some references such as Liu et al., 2014 and Shirasaki et al., 2014 are mentioned in the text but not included in References. Please check them carefully.

We thank the reviewers for pointing out this. We have added the references.